# Urinary metabotypes of newborns with perinatal asphyxia undergoing therapeutic hypothermia

Enrico Valerio[1]*, Veronica Mardegan[1], Matteo Stocchero[1,2], Maria Elena Cavicchiolo[1], Paola Pirillo[1,2], Gabriele Poloniato[1,2], Gianluca D'Onofrio[1], Luca Bonadies[1], Giuseppe Giordano[1,2], Eugenio Baraldi[1,2]

1 Neonatal Intensive Care Unit, Department of Womens' and Childrens' Health, University Hospital of Padua, Padova, Italy, 2 Institute of Pediatric Research (IRP), Fondazione Istituto di Ricerca Pediatrica Città della Speranza, Padova, Italy

* enrico.valerio@aopd.veneto.it

**Data Availability Statement:** We have deposited the data set underlying the results described in the manuscript in the Data Repository of the University

## Abstract

Perinatal asphyxia (PA) still occurs in about three to five per 1,000 deliveries in developed countries; 20% of these infants show hypoxic-ischemic encephalopathy (HIE) on brain magnetic resonance imaging (MRI). The aim of our study was to apply metabolomic analysis to newborns undergoing therapeutic hypothermia (TH) after PA to identify a distinct metabotype associated with the development of HIE on brain MRI. We enrolled 53 infants born at >35 weeks of gestation with PA: 21 of them showed HIE on brain MRI (the "HIE" group), and 32 did not (the "no HIE" group). Urine samples were collected at 24, 48 and 72 hours of TH. Metabolomic data were acquired using high-resolution mass spectrometry and analyzed with univariate and multivariate methods. Considering the first urines collected during TH, untargeted analysis found 111 relevant predictors capable of discriminating between the two groups. Of 35 metabolites showing independent discriminatory power, four have been well characterized: L-alanine, Creatine, L-3-methylhistidine, and L-lysine. The first three relate to cellular energy metabolism; their involvement suggests a multimodal derangement of cellular energy metabolism during PA/HIE. In addition, seven other metabolites with a lower annotation level (proline betaine, L-prolyl-L-phenylalanine, 2-methyl-dodecanedioic acid, S-(2-methylpropionyl)-dihydrolipoamide-E, 2,6 dimethylheptanoyl carnitine, Octanoyl-glucuronide, 19-hydroxyandrost-4-ene-3,17-dione) showed biological consistency with the clinical picture of PA. Moreover, 4 annotated metabolites (L-lysine, L-3-methylhistidine, 2-methyl-dodecanedioic acid, S-(2-methylpropionyl)-dihydrolipoamide-E) retained a significant difference between the "HIE" and "no HIE" groups during all the TH treatment. Our analysis identified a distinct urinary metabotype associated with pathological findings on MRI, and discovered 2 putative markers (L-lysine, L-3-methylhistidine) which may be useful for identifying neonates at risk of developing HIE after PA.

of Padova. Relevant files can be found at this link: https://researchdata.cab.unipd.it/668/.

**Funding:** The authors received no specific funding for this work.

**Competing interests:** The authors have declared that no competing interests exist.

## Introduction

Perinatal asphyxia (PA) is defined as an inadequate oxygen supply to vital organs and peripheral tissues, leading to tissue hypoxia and injury [1]. About three to five per 1,000 deliveries are complicated by PA in developed countries, despite advanced prenatal monitoring and intrapartum obstetric care [2, 3]. Acute PA is followed by a full recovery in most infants, but 20% of patients develop deep brain structure injuries, defining the clinical picture of hypoxic-ischemic encephalopathy (HIE). This condition is burdened by severe early (neonatal death), mid-, and long-term neurological outcomes (cerebral palsy, epilepsy, behavioral disorders, cognitive impairment and neurodevelopmental disability) [2, 4].

Therapeutic hypothermia (TH) is currently the only approved neuroprotective treatment for patients with HIE [5]. It takes effect by reducing vasogenic edema, release of excitatory neurotransmitters, reactive oxygen species (ROS), and pro-inflammatory cytokines, ultimately improving survival in newborns with moderate-to-severe HIE [6]. The "golden window" for initiating TH is in the first 6 hours of life. Given such a short time interval of applicability, much effort is being made to identify additional tools to support the clinical criteria that decide for a patient's cooling [7, 8]. Longitudinal biochemical assessments of ongoing neurological injury in asphyctic patients may help clinicians to decide whether or not to start TH, and to monitor the severity of HIE during patients' hospital stay.

Metabolomics (the latest "-omic" science) can be a valid tool to accomplish these tasks. It consists in a multiparametric approach to studying the complete set of low-molecular-weight (<1 kDa) metabolites in a given biological system by analyzing human fluids like urine, plasma, saliva, and cerebrospinal fluid [9]. This provides a "snapshot" of the metabolic status of an organism as a whole (such as a newborn with PA), at a given point in time. This may pave the way to the discovery of putative metabolites, or sets of metabolites capable of discriminating between the cases that will evolve towards a specific outcome (such as HIE) and those that will not.

No tests are available as yet for an accurate and early diagnosis of HIE. Neonatal metabolic profiling may therefore represent a powerful tool for investigating changes in metabolic pathways relating to perinatal events like PA, and elucidating the mechanisms leading to a higher risk of neurological damage later in life. Metabolomics has been applied to various (mostly non-human) models of PA, with some of the latest studies also extending to human newborns with PA/HIE [10–18].

Our study must be considered a hypothesis-generating study. The aim was to employ untargeted metabolomic analysis to identify dysregulated metabolic processes in newborns undergoing TH after PA, in an effort to establish a distinct metabotype relating to the development of pathological outcomes consistent with HIE on magnetic resonance imaging (MRI). Moreover, the time evolution during TH of the discovered metabotype was investigated to evaluate the robustness of our findings and discover early putative markers of damage.

## Methods

### Study design

We performed a longitudinal monocentric study, enrolling newborns who presented with PA at birth and received TH treatment.

### Patients

The study concerned infants born at >35 weeks of gestation with PA, and undergoing TH according to international guidelines. Patients were admitted to the NICU of the Department

of Women's and Children's Health at the University Hospital in Padova (Italy) from May 2015 to April 2021. The study was approved by the local Ethics Committee (University Hospital of Padua, Padova, Italy, reference 4332/AO/17), and written informed consent was obtained from the parents before patients were enrolled.

Patients were classified by gestational age, birth weight, sex, Apgar score, mode of delivery, need for neonatal resuscitation, type and severity of organ failure, Sarnat score at 60 minutes of life, pH and base excess (BE) derived from cord blood and at 1 hour of life. All patients underwent neurophysiological monitoring with aEEG and video-EEG.

## Sampling

We selected urine as the biofluid to perform metabolomic analysis due to the possibility of non-invasive collection of the samples. Urine samples were collected at 24 (T1), 48 (T2), and 72 (T3) hours of TH. At least 2 ml of urine were collected for each sample by placing a sterile cotton ball inside the newborn's nappy and checking for the presence of urine every 30 minutes. The cotton ball was changed if the neonate did not urinate within 3 hours of its placement, or if it was contaminated with fecal material. After the neonate had urinated, the cotton ball was placed in the barrel of a syringe and squeezed with the plunger to collect the urine it had absorbed in a container prewashed with methanol, for metabolomic analysis. The same brands of nappies and cotton balls were used throughout the study. Samples were stored at -80˚C until analysis.

## Brain MRI grading

All infants included in the study underwent brain MRI at a median age of 6 days of life. The MRI equipment was a Philips Achieva 1.5 Tesla (Philips Healthcare, Best, Netherlands). The MRI sequence protocol included 3D gradient echo T1-weighted images, gradient echo T2-weighted images, turbo spin echo T2-weighted images, and echo planar diffusion-weighted images. MRI scans were blindly reviewed by an experienced pediatric neuroradiologist, and damage was graded according to the score proposed by Barkovich AJ et al. [19]. Patients were scored according to damage of basal ganglia, cortical watershed areas, and sum of basal ganglia and watershed damage, and grouped as: "no HIE" if their cumulative score was 0, or "HIE" if their cumulative score was >0.

## Metabolomic analysis: Sample preparation and UPLC-MS analysis

The analysis was performed at the Mass Spectrometry and Metabolomics Laboratory of the University of Padua's Woman's and Child's Health Department. Urine samples were slowly thawed overnight at +4˚C and then transferred to room temperature for the preparation. Each sample was stirred and centrifuged at 3600 g for 10 min at 10˚C, then 60 μl of the supernatant from each sample were pipetted in a well of 384 wells plate, adding 240 μL of 0.1% formic acid (FA) solution (finale volume 300μL, dilution 1:5). All the procedures for the preparation were automatically managed by a robotic liquid handling system, Multiprobe II Ex (Perkin Elmer, MA, U.S.A).

Untargeted metabolic profiling was performed on an Acquity Ultra Performance Liquid Chromatography (UPLC) system (Waters, U.K.) coupled to a Quadrupole Time-of-Flight (QToF) Synapt G2 HDMS mass spectrometer (Waters MS Technologies, Ltd., U.K.). For mass accuracy, a LockSpray interface was used with a 20 μg/L leucine enkephalin. Data were collected in continuum (profile) mode, in a scanning range of 20–1200 m/z, with acquisition rate of 0.3 s, and with lock mass scans collected every 10 s and averaged over 3 scans for mass correction.

Chromatography was performed using an Acquity HSS T3 (1.7 μm, 2.1 x 100 mm) column (Waters Corporation, U.S.A.) kept at 50°C. The flow rate of the mobile phase was set at 0.5 ml/min, and each sample run lasted 12 min, with 5 μl of the sample injected for each run. The gradient mobile phase consisted of water with 0.1% FA (A) and methanol with acetonitrile in a 90:10 ratio with 0.1% FA (B). Each sample run lasted 11 min of an isocratic phase of 5% B for 1 min, a linear increase to 30% B in 2.5 min, a linear increase to 95% B in 3 min, an isocratic phase of 95% B for 1.5 min, a washout phase of 5% B for 3 min.

The electrospray source of QToF was operated in positive (ESI+) and negative (ESI-) ionization mode with a capillary voltage set at 3 kV and 1.5 kV, respectively. We acquired data also in $MS^E$ scan mode. This last is a non-selective fragmentation technique based on the passage of ionized molecules through the collision cell at high and low collision energy to obtain structural information of the molecules.

Quality Control samples (QCs) and Standards Solution Samples (Mixes) were used to assess reproducibility and accuracy during the analysis, and examine the metabolite content of the samples. The QCs were prepared from an aliquot (50 μL) of each sample, pooled together and diluted with three different dilution factors (1:3, 1:5, 1:7) with 0.1% FA solution in water, treated as the samples. The Mix consisted of nine compounds of known exact mass and retention time. The QCs and Mixes were injected at regular intervals of 15 samples during the sequence, together with blank samples, to identify specific ions from the mobile phase, and any contaminants.

The sample injection order was randomized in order and prevent any spurious classification deriving from the position of the sample in the sequence.

## Data pre-processing and pre-treatment

Data were pre-processed using Progenesis QI software (Waters Corporation, U.S.A.). The parameters used for data extraction were optimized through the preliminary processing of the QCs. Specifically, 0.5 was set as filter to import the raw data, and the QC in the middle of the sequence was selected as reference for the automatic retention time alignment of the samples in the sequence. The sensitivity of the automatic algorithm for peak picking was set at 5 in the time range from 0.4 to 8.0 min. As a result, the so-called Rt_mass variables (where "Rt" is the retention time and "mass" is the mass to charge ratio m/z of the spectral feature) were generated.

Features with more than 10% of missing data were eliminated. For each variable passing such a filter, missing data were imputed with a random number between zero and the minimum value measured for that variable. Data were calibrated on the basis of the local linear regression models obtained considering the trend of the QCs with the run order as explained in Santamaria et al., 2019 [20]. Probabilistic quotient normalization was applied to take into account dilution effects. Variables with a coefficient of variation greater than 20% in the QCs were excluded. Data were log-transformed and mean-centred prior to performing data analysis.

## Statistical data analysis

Data representing the characteristics of the recruited newborns were analysed by t-test or Mann-Whitney test for continuous normally or non-normally distributed data, respectively, and by Chi-squared test for categorical variables. Normality of the data was assessed by Shapiro-Wilk test (p>0.10).

Multivariate data analysis was applied to investigate the metabolomic data. Specifically, Principal Component Analysis (PCA) was used for outlier detection and PLS for classification

(PLS2C) with stability selection for discovering a distinct set of variables able to distinguish between "HIE" and "no HIE" patients. PLS2C has been recently introduced to solve the general G-class problem [21], when correlation, redundancy and noise affect the data. It overcomes the weakness of the theoretical basis of PLS-DA and is suitable for metabolomic investigations. Specifically, the relationships between observations can be discovered exploring the score space of the model and relevant variables highlighted by stability selection. The idea underlying stability selection is that real differences should be present consistently, and therefore should be found even under perturbation of the data by sub-sampling. Bootstrap has been applied to extract 200 sub-samples from the training set and, thus, PLS2C with variable selection based on variable influence on projection (VIP) was applied to each sub-sample to obtain 200 classification models, each one based on a different optimal subset of variables. For each variable, the number of models whose optimal subset included that variable was considered to calculate the so called "relevance score" that was used to select the relevant variables. The optimal subset was determined on the basis of the maximum Matthews Correlation Coefficient calculated by 5-fold cross-validation.

The set of the selected relevant variables was investigated by univariate data analysis to discover if single variables behave as putative markers of HIE and to study their evolution during TH. Specifically, the Volcano plot built considering the p-value of the Mann-Whitney test and the fold change between "HIE" and "no HIE" group was investigated to discover putative markers of HIE, and the time evolution during hypothermia was studied applying Linear Mixed Effects (LME) modelling for longitudinal data [22]. In LME modelling, each relevant variable was modelled including the effects of time and group assuming that the growth curves show a similar linear pattern across patients, but that important individual differences may be exhibited, both in intercept and in slope, as typically occurs with longitudinal data.

Data analysis was performed using in-house R-functions implemented by R 4.0.4 platform (R Foundation for Statistical Computing).

### Variable annotation

The relevant variables selected by data analysis were annotated by searching our in-house database of commercial standards, the METLIN metabolite database and the Human Metabolome Database using a different level of confidence [23].

Specifically, annotation was based on accurate mass, retention time and fragmentation patterns, where available. Annotation level 1 was assigned to compounds with a difference of ≤10ppm for m/z, and 0.2 min for rt, with respect to standards of our in-house database, which were analyzed under identical conditions to the current analysis. Level 2 was attributed to metabolites with m/z ≤10ppm and with similar fragmentation patterns with respect to the online databases, whereas level 3 was assigned to compounds with m/z ≤10ppm on the online databases. Level 4 was used for repeatable signals of mass spectrum, with no annotation in the databases used.

## Results

### Characteristics of the recruited patients

A total of 53 neonates were recruited: 21 showed signs of neurological damage on MRI (the "HIE" group), and 32 did not (the "no HIE" group). Table 1 shows the characteristics of the two groups. No significant differences emerged between them in terms of the patients' clinical data or pharmacological treatment, adopting a significance level of α = 0.05. Due to practical reasons related to the fact that asphyctic newborns do not always urinate regularly, it was not possible to collect samples at each time point for every neonate. For only 9 neonates (6

**Table 1. Characteristics of the recruited patients.**

| | "no HIE" group N = 32 | "HIE" group N = 21 | p |
|---|---|---|---|
| Sex, male (female) | 18 (14) | 14 (7) | 0.57 |
| Gestational age [days] | 275 (12) | 273 (11) | 0.33 |
| Birth weight [grams] | 3292 (565) | 3270 (575) | 0.21 |
| Delivery mode, vaginal (caesarean section) | 20 (12) | 11 (9) | 0.77 |
| SARNAT 60 min | 2 [2,2] | 2 [2;2] | 0.62 |
| Apgar 1 min | 2.5 [2;4] | 3 [1;5] | 0.91 |
| Apgar 5 min | 5 [4;7] | 5 [3;7] | 0.69 |
| Apgar 10 min | 7 [5;8] | 6.5 [4.0;7.3] | 0.31 |
| Hypoglycemia at birth, yes (no) | 9 (23) | 5 (16) | 1.00 |
| pH | 7.00 (0.15) | 6.99 (0.14) | 0.20 |
| BE | -14.9 [-17.8;-10.9] | -16.6 [-20.6;-11.3] | 0.78 |
| pH at 1 h | 7.14 [7.06;7.22] | 7.09 [7.03;7.23] | 0.87 |
| EB at 1 h | -17.0 [-21.2;-12.4] | -18.3 [-22.0;-11.3] | 0.76 |
| Early-onset sepsis, yes (no) | 17 (15) | 11 (10) | 1.00 |
| Late-onset sepsis, yes (no) | 5 (27) | 6 (15) | 0.31 |
| Ventilated at birth, yes (no) | 30 (2) | 20 (1) | 1.00 |
| Thoracic compressions at birth, yes (no) | 7 (25) | 3 (18) | 0.72 |
| Drug resuscitation at birth, yes (no) | 8 (24) | 4 (17) | 0.74 |
| Antibiotic therapy, yes (no) | 32 (0) | 21 (0) | 1.00 |
| Inotropes, yes (no) | 9 (23) | 5 (16) | 1.00 |
| Phenobarbital, yes (no) | 15 (17) | 12 (9) | 0.58 |
| Phenytoin, yes (no) | 0 (32) | 2 (19) | 0.15 |
| Benzodiazepines, yes (no) | 1 (31) | 2 (19) | 0.56 |
| Other antiepileptic drugs, yes (no) | 1 (31) | 0 (21) | 1.00 |
| RBCs, yes (no) | 5 (27) | 7 (14) | 0.18 |
| FFP, yes (no) | 20 (12) | 15 (6) | 0.56 |
| Platelet transfusion, yes (no) | 8 (24) | 3 (18) | 0.49 |
| Mild/moderate AKI*, yes (no) | 12 (20) | 10 (11) | 0.57 |
| Severe AKI$, yes (no) | 0 (32) | 2 (19) | 0.15 |
| Liver enzymes elevation, yes (no) | 19 (13) | 16 (5) | 0.25 |
| Troponin elevation, yes (no) | 16 (16) | 13 (8) | 0.42 |
| Cardiac failure, yes (no) | 9 (23) | 5 (16) | 1.00 |
| Coagulopathy, yes (no) | 18 (14) | 11 (10) | 1.00 |

Normally distributed data are reported as means (SD), non-normally distributed data as medians [25th-75th], and categorical variables as numbers of occurrences. RBCs, red blood cells; FFP, fresh frozen plasma; AKI, acute kidney injury.

*, defined as a rise in blood urea and/or creatinine with no reduction in urinary output

$, defined as a rise in blood urea and/or creatinine with a reduction in urinary output +/- peritoneal dialysis.

belonging to "no HIE" group and 3 belonging to "HIE" group) at least one urine sample was collected at each time point. Moreover, a total of 18 urine samples were collected at T1, 44 at T2, and 34 at T3 during TH.

## Metabolomic investigation

A dataset including 1118 Rt_mass variables was generated from the raw data, merging the two datasets obtained with the positive and negative ionization mode. No outliers were detected by PCA on the basis of the T2 and Q tests, setting a significance level of $\alpha = 0.05$. The first

available urine sample during TH was used to compare the urinary metabolome of the two groups of patients in order to discover urinary metabotypes associated to HIE. Specifically, 17 samples were collected at T1, 29 at T2 and 7 at T3. No significant differences came to light between the two groups in the distribution of the samples at each time point (chi-squared test p = 0.14).

A subset of 111 relevant variables was pinpointed using PLS for classification and stability selection, adopting a significance level of α = 0.05. Considering the relevant variables, the PLS2C model obtained showed two components, a Matthews Correlation Coefficient in calculation (MCC) equal to 0.73 (p = 0.05), a MCC calculated by means of 21 repeated 5-fold cross-validations (MCCcv) of 0.40 (p = 0.008) and an out-of-the-bag MCC (MCCoob) of 0.37. The model passed the permutation test on the class response (500 random permutations). It was consequently judged reliable and the metabolic signature discovered by PLS for classification could be assumed to be suitable for distinguishing between patients with and without HIE. The scores obtained with the model were used to obtain the scatter plot shown in Fig 1: the circles representing samples of patients belonging to the same group are close together, and the two groups occupy two different regions of the plot.

The behavior of the 111 relevant predictors was investigated with a Volcano plot and linear mixed effects (LME) modelling. Fig 2 summarizes the results of the data analysis.

The Volcano plot (Fig 2A) illustrates the ability to distinguish between the "HIE" and "no HIE" groups using the first urine samples becoming available for the recruited patients. A set of 35 variables showing a fold change greater than 2 (i.e. absolute $\log_2$(FC) greater than 1) or significantly different medians (α = 0.05) was selected as interesting variables responsible of the differences between "HIE" and "no HIE" group. This set can be seen as the metabotype associated with pathological outcomes of PA on brain MRI. Sixteen of these 35 variables were annotated (Table 2). It is important to remark that the metabotype has been discovered by a multivariate approach and that its components may not behave as single markers because the differences between "HIE" and "no HIE" group are due to their cooperative contribution.

The scatter plot of Fig 2B summarizes the results of the LME analysis performed to study the longitudinal trend of the selected metabolites. Since data for all the three time points were available for only nine patients, the analysis should be considered explorative and the trends discovered should be used only for hypothesis generation because the study is underpowered to detect the real effects of time and group. Specifically, 22 metabolites changed linearly during time (16 increased and 6 decreased during TH) and six showed differences due to groups. Interestingly, four annotated metabolites (L-lysine, L-3-methylhistidine, 2-methyl-dodecanedioic acid, S-(2-methylpropionyl)-dihydrolipoamide-E) retained a significant difference between the "HIE" and "no HIE" groups during TH independently of time (Fig 3) whereas for three annotated metabolites the differences observed between the two groups disappeared during time.

## Discussion

In this study we examined dynamic perturbations in the urinary metabolome in the setting of PA, analyzing urinary samples collected over the first three days of life from asphyctic infants undergoing TH. Our untargeted metabolomic analysis found a total of 111 relevant predictors that, taken together, had the power to discriminate between our "HIE" and "no HIE" groups. Thirty-five of these metabolites independently showed such a discriminatory power. Among them, there were four (L-alanine, Creatine, L-3-methylhistidine, and L-lysine; see Table 2) reaching Annotation level 1 (meaning strong confidence in metabolite identification). L-3-methylhistidine and L-lysine also maintained a significant difference between the "HIE" and "no HIE" groups throughout the hypothermia treatment (see Fig 3).

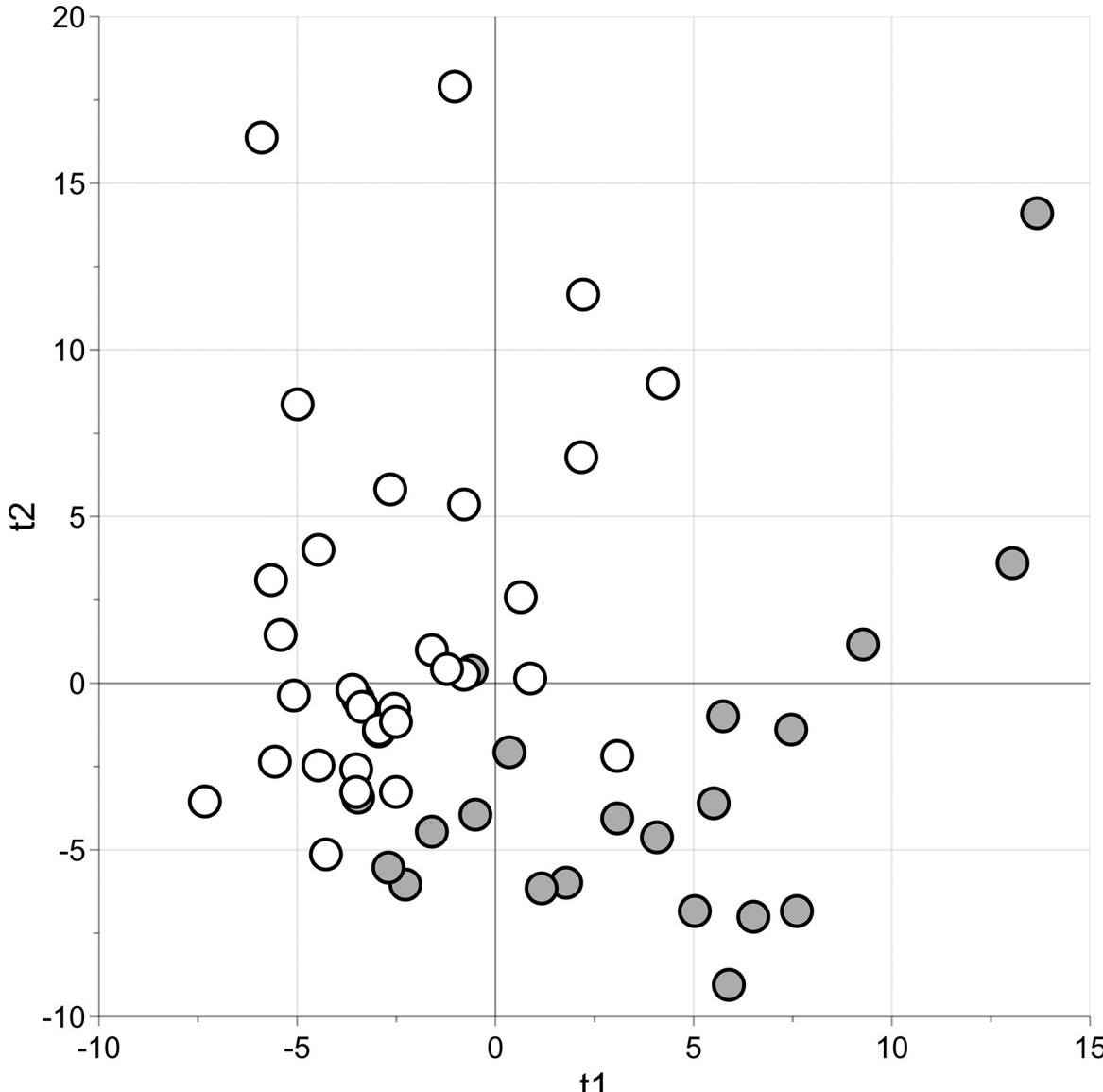

**Fig 1. PLS for classification.** Score scatter plot; grey circles represent urine samples of "HIE" patients, and white circles those of "no HIE" patients.

The prognostic potential of these four metabolites lies in that each one of them is capable, during TH, of distinguishing between infants who will develop HIE and those who will not. L-alanine, Creatine, and L-3-methylhistidine all belong to the class of compounds related to energy metabolism. Their involvement in different (cytoplasmic and intramitochondrial) cellular energy-producing pathways suggests a profound and multimodal derangement of cellular energy metabolism following PA.

Alanine is a non-essential amino acid, synthesized from pyruvic acid. Given its close biochemical relationship with pyruvate, alanine is closely connected to all major cellular metabolic energy pathways, such as the tricarboxylic acid cycle (TCA), gluconeogenesis, and the Cahill (or glucose-alanine) cycle [24–26]. Urinary alanine was found higher in the "HIE" group than in the "no HIE" group, consistently with a shift towards anaerobic metabolism in

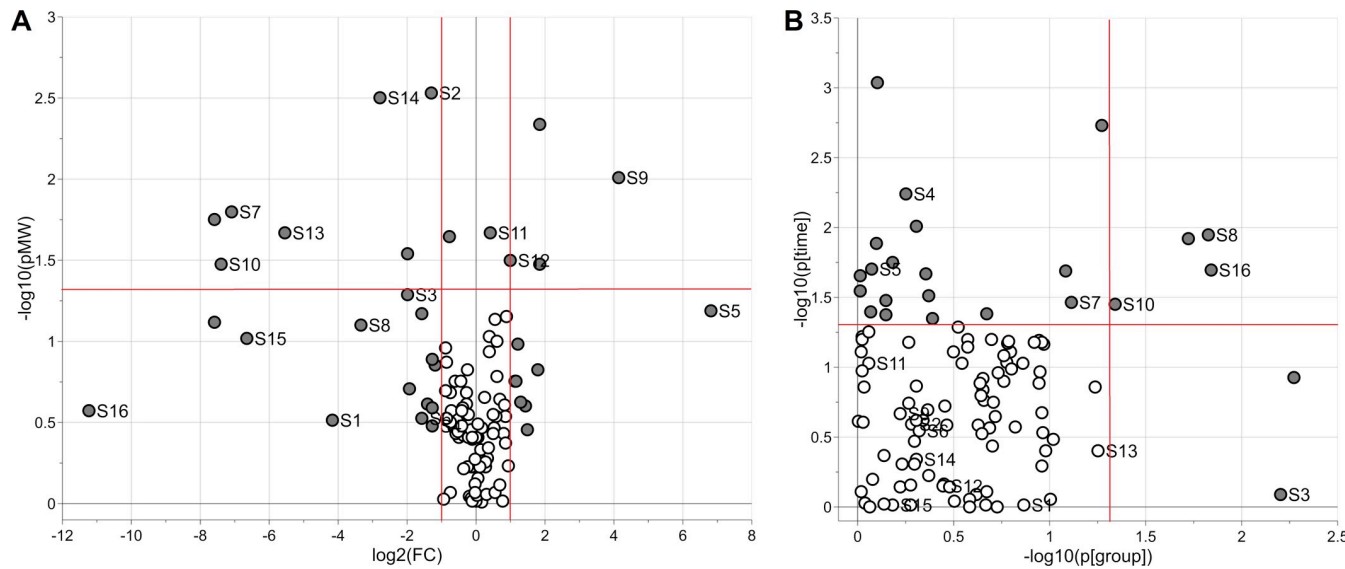

**Fig 2. Analysis of the metabolic signature discovered by PLS for classification.** The Volcano plot (panel A) shows the negative logarithm of the p-value of the Mann-Whitney test (vertical axis) versus the logarithm of the fold change calculated as the ratio between the median of the "HIE" group and the median of the "no HIE" group (horizontal axis). Panel B shows the negative logarithm of the p-value of time (vertical axis) and the negative logarithm of the p-value of the factor group with levels "HIE" and "no HIE" (horizontal axis) obtained by LME modelling. Metabolites showing fold change greater than 2 or p-value<0.05 are reported as grey circles. Thresholds at the significance level α = 0.05, and the limits corresponding to a 2-fold change are shown as red lines. In figure, annotated metabolites are indicated as follows: S1 = 19-hydroxyandrost-4-ene-3,17-dione, S2 = 2,6 dimethylheptanoyl carnitine, S3 = 2-methyl-dodecanedioic acid, S4 = 3-mercaptolactate-cysteine disulfide, S5 = Creatine, S6 = Dihydrostreptomycin 6-phosphate, S7 = Glycylprolylhydroxyproline, S8 = L-3-methylhistidine, S9 = L-alanine, S10 = L-lysine, S11 = L-prolyl-L-phenylalanine, S12 = N-methylethanolamine phosphate, S13 = Norcotinine, S14 = Octanoylglucuronide, S15 = Proline betaine, S16 = S-(2-methylpropionyl)-dihydrolipoamide-E.

**Table 2. Annotated variables of the metabolic signature discovered by PLS for classification.**

| Annotation | Level | m/z | Rt | relevance score | pMW | $log_2(FC)$ | p[time] | p[group] | coeff[time] |
|---|---|---|---|---|---|---|---|---|---|
| L-alanine | 1 | 90.0561 | 0.624 | 0.995 | 0.010 | 4.138 | 0.214 | 0.598 | |
| L-lysine | 1 | 130.0873 | 0.523 | 0.985 | 0.033 | -7.403 | 0.035 | 0.046 | 0.49 |
| Creatine | 1 | 132.0749 | 0.624 | 0.97 | 0.065 | 6.805 | 0.02 | 0.848 | 0.51 |
| Proline betaine | 3 | 144.1029 | 0.661 | 0.88 | 0.096 | -6.65 | 0.966 | 0.656 | |
| N-methylethanolamine phosphate | 3 | 156.0426 | 0.575 | 0.945 | 0.032 | 0.979 | 0.707 | 0.36 | |
| Norcotinine | 3 | 163.0876 | 3.776 | 0.995 | 0.022 | -5.535 | 0.398 | 0.056 | |
| L-3-methylhistidine | 1 | 170.0935 | 0.58 | 0.8 | 0.079 | -3.328 | 0.011 | 0.015 | 0.51 |
| L-prolyl-L-phenylalanine | 3 | 227.1189 | 3.573 | 0.655 | 0.022 | 0.407 | 0.094 | 0.871 | |
| 2-methyl-dodecanedioic acid | 3 | 227.1628 | 6.454 | 0.975 | 0.052 | -1.994 | 0.821 | 0.006 | |
| 3-mercaptolactate-cysteine disulfide | 3 | 242.0163 | 0.715 | 0.705 | 0.334 | -1.27 | 0.006 | 0.56 | -0.94 |
| Glycylprolylhydroxyproline | 3 | 286.1411 | 0.71 | 0.94 | 0.016 | -7.092 | 0.034 | 0.077 | 0.55 |
| S-(2-methylpropionyl)-dihydrolipoamide-E | 3 | 300.1091 | 1.207 | 0.765 | 0.266 | -11.227 | 0.02 | 0.014 | 1.16 |
| 2,6 dimethylheptanoyl carnitine | 3 | 302.2336 | 5.561 | 0.57 | 0.003 | -1.303 | 0.254 | 0.524 | |
| 19-hydroxyandrost-4-ene-3,17-dione | 3 | 303.1963 | 4.294 | 0.85 | 0.307 | -4.169 | 0.967 | 0.137 | |
| Octanoylglucuronide | 3 | 343.1371 | 5.567 | 0.995 | 0.003 | -2.794 | 0.453 | 0.497 | |
| Dihydrostreptomycin 6-phosphate | 3 | 686.2366 | 6.112 | 0.57 | 0.299 | -1.568 | 0.284 | 0.479 | |

Level indicates the annotation level [23], the relevance score is the relevance score calculated by PLS2C with stability selection, pMW is the p-value of the Mann-Whitney test, FC is the fold change, p[time] is the p-value for the factor time calculated by LME analysis, p[group] is the p-value for the factor group calculated by LME analysis, and coeff[time] is the coefficient of the fixed effect of time.

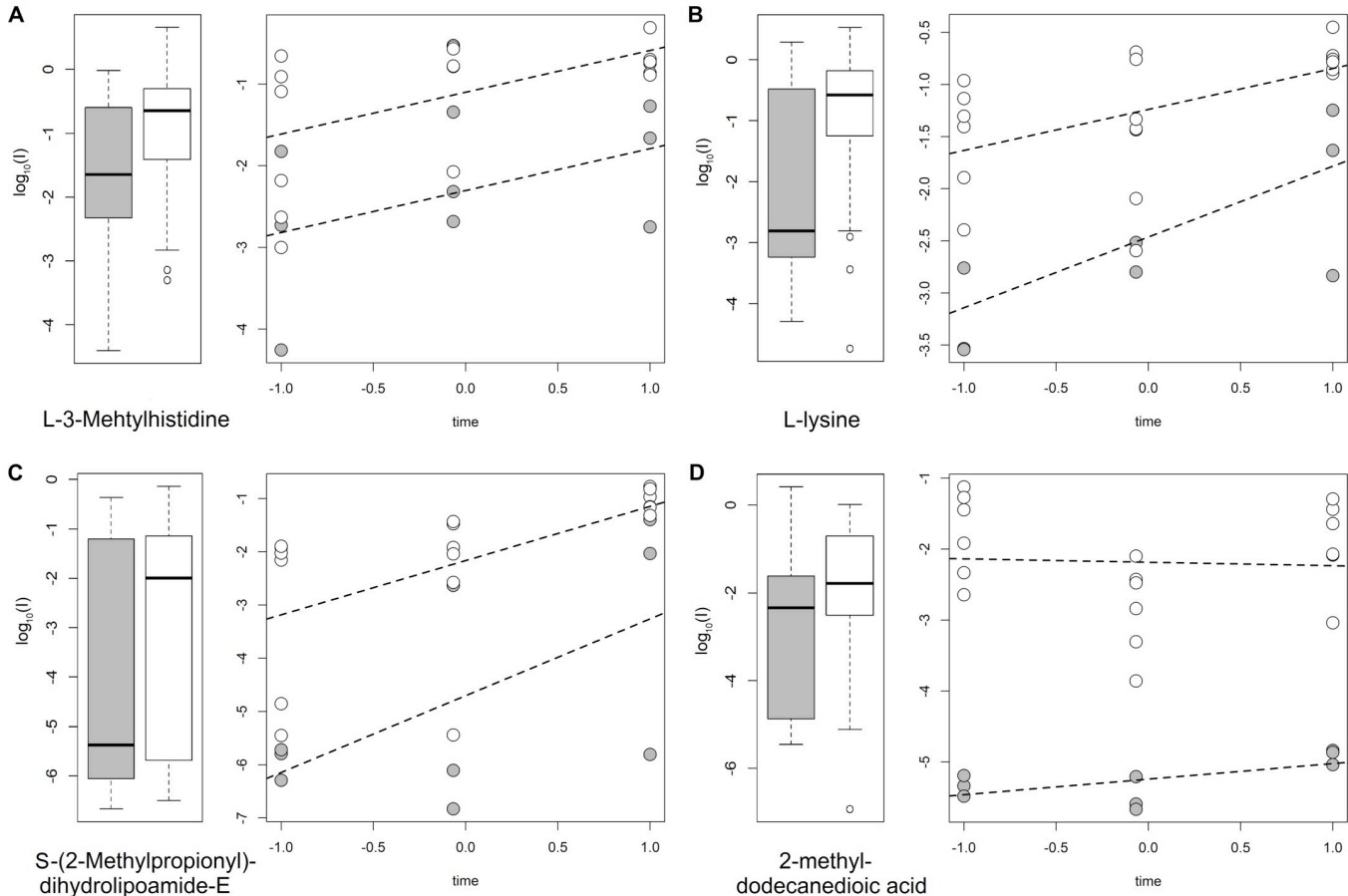

**Fig 3. Boxplots showing the distribution of the relevant metabolites considering the first collected urine samples and the trends of the same features during TH, as obtained from the fixed effects of the LME models; grey is used for "HIE" patients, and white for "no HIE" patients.**

asphyctic infants with more severe prognosis. A similar rise in urinary alanine concentrations in newborns with HIE was reported by Piñeiro-Ramos et al. (2022) [27]. O'Boyle et al. (2021) [25], and Stacey et al. (2013) [15] also found a higher alanine content in the cord blood of infants with PA/HIE compared with healthy controls.

Creatine is an endogenous metabolite originating from arginine and glycine in a two-step process involving the kidney and liver. Endogenous creatine (and creatine from dietary intake) is exported from the liver and internalized by tissues with a high energy demand, mainly skeletal muscle, heart, kidney, small intestine, and brain [28]. It can be *phosphorylated* in these tissues *and serve as an "energy buffer" to restore local levels of ATP, when the intracellular energy charge (ATP:ADP ratio) is low [29]*. We noted higher urinary creatine concentrations in the "HIE" than in the "non-HIE" group. This can be interpreted either as the result of an enhanced rate of creatine mobilization from the liver to tissues lacking in energy in an attempt to restore their energy levels, or as a consequence of suffering and possible cell necrosis of the most energy-dependent tissues (i.e., brain, heart, skeletal muscle, kidney) in severely asphyctic infants. Our results are consistent with those of Longini et al. (2015) [11] and Locci et al. (2018) [13].

L-3-methylhistidine is derived from the methylation of human actin and myosin peptide chains, and is released on the breakdown of these muscle proteins. A high urinary content of L-3-methylhistidine reflects a failure of its reutilization following its release [30]. We found

that L-3-methylhistidine maintained its ability to discriminate between the "HIE" and the "no HIE" groups throughout 72 hours of TH, with concentrations always higher in the "no HIE" group. A possible explanation for this may come from animal and human models in which L-3-methylhistidine was examined as a marker of kidney injury: the rate of its renal excretion into urine was slower the greater the glomerular filtration impairment [31, 32]. Urinary L-3-methylhistidine levels in newborns with HIE could therefore reflect the severity of their glomerular filtration impairment.

Urinary lysine was found higher in the "no HIE" than in the "HIE" group. As discussed before for L-3-methylhistidine, this discriminatory capability persisted throughout the TH process (see Fig 3). Lysine is an essential amino acid expressed particularly in neuronal proteins, and it is directly involved in modulating gene expression [33]. Experimental studies on murine models also suggest that this amino acid has a neuroprotective [34] and anti-inflammatory role. In their study on urinary metabolome modifications in newborns with moderate-to-severe HIE undergoing TH, Piñeiro-Ramos et al. (2022) [27] found higher levels of lysine degradation products in the group with brain injury, again indirectly suggesting a neuroprotective role of the intact amino acid.

Among the compounds with a lower annotation level, seven (proline betaine, L-prolyl-L-phenylalanine, 2-methyl-dodecanedioic acid, S-(2-methylpropionyl)-dihydrolipoamide-E, 2,6 dimethylheptanoyl carnitine, Octanoylglucuronide, 19-hydroxyandrost-4-ene-3,17-dione) are in any case worth a brief discussion, given their biological consistency with the clinical picture of PA.

Proline betaine concentrations in human blood are rather variable, ranging from undetectable to about 50 mmol/l. Its urinary excretion reflects plasma concentrations [35].

There are some evidences that proline betaine, like its related metabolite glycine betaine, might exert an osmoprotective role for the kidney; [36] its relative deficiency, evident in the "HIE" compared to the "no HIE" patients of our series, might contribute to the renal failure often accompanying severe cases of PA.

Prolylphenylalanine is a dipeptide composed of proline and phenylalanine. It is an incomplete breakdown product of protein digestion or protein catabolism [37].

Its presence in urines of our patients, consistently with the urinary presence of L-3-methylhistidine, further remarks the intensity of the catabolic state occurring in a condition like asphyxia.

2-methyl-dodecanedioic acid belongs to the family of dicarboxylic acids, which are water-soluble and have a metabolic pathway intermediate between those of lipids and carbohydrates [38].

Under normal conditions, urinary excretion of this compound is low, since mitochondrial oxydation is rather efficient; [39] its slight increase in urines of the "HIE" group over time, as depicted in Fig 3, may suggest a blockade of lipid oxydation in such patients, with consequent cellular energy failure.

S-(2-methylpropionyl)-dihydrolipoamide-E belongs to the class of organic compounds known as fatty amides. These are carboxylic acid amide derivatives of fatty acids, that are formed from a fatty acid and an amine.

Specifically, S-(2-methylpropionyl)-dihydrolipoamide-E is an intermediate in valine, leucine and isoleucine degradation. It is the second to last step in the synthesis of branched chain fatty acid. In our series, this metabolite was significantly diminished in the "HIE" compared to the "no HIE" group. Consistently with these results, its diminution has been found significantly associated with alteration in glutamate levels in blood of patients that were at risk of thrombotic stroke and later diagnosed with stroke, compared to healthy subjects [40].

2,6 dimethylheptanoyl carnitine is a medium-chain acylcarnitine, normally represented in urines of healthy subjects [41].

Accumulation over time of acylcarnitines and other acyl derivatives in urinary samples has previously been described both in animal models of PA and in asphyctic infants [14, 18, 27], and is possibly related to the inefficiency of the β-oxidation process under conditions of disrupted cellular energy metabolism like those characterizing PA.

Octanoylglucuronide is a glucuronated medium-chain fatty acid that has been found in urines of human subjects with congenital errors of metabolism leading to a defective oxidation of medium-chain tryglicerides [42]. Similarly to 2,6 dimethylheptanoyl carnitine, urinary accumulation of octanoylglucuronide might reflect the impairment of mitochondrial fatty acid oxidation in asphyxiated infants.

19-Hydroxyandrost-4-ene-3,17-dione, also known as 19-HAED, belongs to the class of androgens and derivatives. It is normally secreted from the adrenal glands both in males and females, and is also produced by the placenta. During normal gestation, its levels in serum concentration of maternal venous blood show a constant increase, reaching a maximum at delivery [43].

Steroids exert a number of protective activities over fetal central nervous system, and are pivotal for neuronal and glial growth and maturation in the fetal brain [44, 45].

The relatively low presence of a neuroprotective steroid like 19-HAED in the "HIE" compared to the "no HIE" infants in our series is consistent with the brain damage in the former group of patients.

It is worth noting that most of the relevant features only distinguished between the "HIE" and "no HIE" groups in the first phase of TH, and then the differences between the two groups gradually faded. This could be due both to the neuroprotective effect of TH and (in animal models at least) to endogenous mechanisms of damage repair [46], consistently with previous observations [27]. In contrast with this general trend, L-lysine, L-3-methylhistidine, 2-methyl-dodecanedioic acid and S-(2-methylpropionyl)-dihydrolipoamide-E maintained a significant difference between the two groups throughout the hypothermic treatment, proving that a part of the metabotype marking the damage is preserved during TH.

## Our study has some limitations

We selected urines rather than plasma as a biofluid to perform metabolomic analysis, due to the possibility of non-invasive collection of the samples. Asphyctic newborns do not always urinate regularly, and this was the principal reason for the low number of patients having urines collected at all time points.

Given the small number of subjects, we were unable to subgroup our HIE patients by severity on brain MRI due to the small numbers involved. Distinguishing only between patients with and without HIE (whatever the severity of the damage involved in the former) nonetheless enabled us to characterize a set of metabolites capable of discriminating newborns with even mild HIE on MRI, and—from the clinician's point of view–it is undoubtedly worth initiating TH in these infants.

Finally, the longitudinal part of our analysis (ie, the investigation of the evolution during TH of the discovered metabotype) was underpowered because based on only nine patients (those with available data for all the three time points). Then, this part of the data analysis should be considered explorative and the trends discovered should be used only for hypothesis generation.

In conclusion, metabolomics identified distinct urinary metabotypes associated with pathological outcomes on MRI in newborns with PA treated with TH. The more pronounced metabolic derangements related to cellular energy metabolism. Specifically, L-lysine and L-3-methylhistidine maintained a significant difference during the whole hypothermic treatment

between patients who will develop HIE and those who will not, and can be considered putative markers for the early diagnosis of HIE. Our findings pave the way to develop rapid chemical test that may facilitate the early identification of neonates with PA at risk of developing brain injury. Further investigations will be dedicated to validate our results in larger datasets to ascertain whether these changes are specific to HIE, to quantify the power in prediction of the putative markers and to better understand the bio-chemical mechanisms underlying TH.

## Author Contributions

**Conceptualization:** Veronica Mardegan, Matteo Stocchero, Paola Pirillo, Giuseppe Giordano, Eugenio Baraldi.

**Data curation:** Enrico Valerio, Veronica Mardegan, Gianluca D'Onofrio, Giuseppe Giordano.

**Formal analysis:** Matteo Stocchero, Giuseppe Giordano, Eugenio Baraldi.

**Investigation:** Veronica Mardegan, Gianluca D'Onofrio.

**Methodology:** Enrico Valerio, Veronica Mardegan, Matteo Stocchero, Paola Pirillo, Gabriele Poloniato, Luca Bonadies, Giuseppe Giordano, Eugenio Baraldi.

**Project administration:** Paola Pirillo, Gabriele Poloniato, Gianluca D'Onofrio, Eugenio Baraldi.

**Resources:** Gabriele Poloniato, Giuseppe Giordano, Eugenio Baraldi.

**Software:** Matteo Stocchero, Paola Pirillo, Gabriele Poloniato.

**Supervision:** Maria Elena Cavicchiolo, Gabriele Poloniato, Giuseppe Giordano, Eugenio Baraldi.

**Validation:** Enrico Valerio, Matteo Stocchero, Paola Pirillo, Gabriele Poloniato, Gianluca D'Onofrio.

**Visualization:** Eugenio Baraldi.

**Writing – original draft:** Enrico Valerio.

**Writing – review & editing:** Enrico Valerio, Veronica Mardegan, Matteo Stocchero, Maria Elena Cavicchiolo, Paola Pirillo, Gabriele Poloniato, Gianluca D'Onofrio, Luca Bonadies, Giuseppe Giordano, Eugenio Baraldi.

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
