## [Decision Letter · Decision Letter 0]

3 Jun 2022

PONE-D-22-12491Urinary metabotypes of newborns with perinatal asphyxia undergoing therapeutic hypothermiaPLOS ONE

Dear Dr. Valerio,

Thank you for submitting your manuscript to PLOS ONE. After careful consideration, we feel that it has merit but does not fully meet PLOS ONE’s publication criteria as it currently stands. Therefore, we invite you to submit a revised version of the manuscript that addresses the points raised during the review process.

In particular, the authors should address the following methodological points: 1, The results related to the longitudinal part of the study are not described; 2, the longitudinal effect due to the therapy has not been shown; 3, the rationale behind the selection of urine to describe hypoxia should be included; 4, the description of levels 1, 2 and 3 should be included; 5, a comment on the “golden window” of 6 hours and on how fast is the procedure should be added. Furthermore, figures 2 and 3 must be improved. The potential role of some of the found metabolites should be stated. The discussion on selected metabolites and on their clinical relevance should be improved.

We look forward to receiving your revised manuscript.

Kind regards,

Andrea Motta

Academic Editor

PLOS ONE

Journal Requirements:

" ext-link-type="uri" xlink:type="simple">https://journals.plos.org/plosone/s/file?id=ba62/PLOSOne_formatting_sample_title_authors_affiliations.pdf"

2. Please provide additional details regarding ethical approval in the body of your manuscript. In the Methods section, please ensure that you have specified the name of the IRB/ethics committee that approved your study.

Reviewers' comments:

Reviewer's Responses to Questions

**Comments to the Author**

1. Is the manuscript technically sound, and do the data support the conclusions?

Reviewer #1: Yes

Reviewer #2: Yes

2. Has the statistical analysis been performed appropriately and rigorously? 

Reviewer #1: Yes

Reviewer #2: Yes

3. Have the authors made all data underlying the findings in their manuscript fully available?

Reviewer #1: No

Reviewer #2: No

4. Is the manuscript presented in an intelligible fashion and written in standard English?

Reviewer #1: Yes

Reviewer #2: Yes

5. Review Comments to the Author

Reviewer #1: This manuscript describes the longitudinal metabolomics analysis in newborn undergoing to therapeutic hypothermia with the objective of finding metabolic signatures for diagnostic purpose. On the technical point of view, the paper sounds, however this reviewer has some comments that needed to be addressed.

• The results related to the longitudinal part of the study have not been described at all. The paper claims to investigate metabotypes of newborn undergoing to therapeutic hypothermia, but the longitudinal effect due to the therapy on the metabolic signature has not been showed. If there are no results on the longitudinal part of the study, would be more appropriate to talk about metabolic signature of perinatal asphyxia in newborn.

• I understand that collecting blood for newborn is a quite an invasive procedure. However, in the limitations of the study as well as in the text, it should be explained the rationale behind selection of urine as samples, and the limitations of these sample to investigate perinatal hypoxia.

• Figure 2 and 3 must be improved adding legends, name of metabolites and experimental conditions. In the way they are presented, without no information at all inside the figures, they are not so informative and also difficult to read. Probably a table would be more informative. For instance the volcano plots without name of metabolites is not really useful.

• A small discussion/hypothesis of the potential role of some of these metabolites identified with level 3 would be appropriate.

• I do not understand the rationale behind selection of metabolites showed in figure 3. Two of these metabolites were not even mentioned in the discussion (see previous comment).

Reviewer #2: The manuscript is about a small scale study on infants born with Perinatal Asphyxia w/wo hypoxic-ischemic encephalopathy with the limits declared by the authors themselves. LCMS analysis followed by statistical analysis allowed to identify 111 predictors and 35 metabolites showed independent discriminatory power. Of these, only four were identified ("level 1") while other 12 were annotated with a putatively less accurate "level 3". However, it is not described what this level 3 exactly means and on which basis the group of these 12 metabolites was annotated. Furthermore, two of them have been considered 'relevant metabolites' and reported in Figure 3 together with "level 1" L-Lysine and L-3-methyl histidine, i.e. 2-methyl-dodecanedioc acid and S-(2-methylpropionyl)-dihydrolipoamide E. Yet, no mention was made about these latter two metabolites supporting the observed clinical effect in reason of their putative identification. Please improve the discussion including these aspects.

A final consideration: if the golden window for initiating therapeutic hypothermia is in the first 6 hours of the child's life, is it realistic to imagine that in this time interval it is possible to collect the urine, carry out this kind of analysis and achieve timely results?

Minor issues:

Pg13 L 274 "Sixteen of these 35 variables" and not seventeen, according to Table 2

6. PLOS authors have the option to publish the peer review history of their article (what does this mean?). If published, this will include your full peer review and any attached files.

Reviewer #1: No

Reviewer #2: No

---

## [Author Response · Author response to Decision Letter 0]

14 Jul 2022

Padova, July 14th, 2022

To the Reviewers

We thank the Reviewers for their helpful and constructive comments on the manuscript; we have it revised accordingly to their indications. Here follows a point-by-point response to the issues raised:

Reviewer #1

This manuscript describes the longitudinal metabolomics analysis in newborn undergoing to therapeutic hypothermia with the objective of finding metabolic signatures for diagnostic purpose. On the technical point of view, the paper sounds, however this reviewer has some comments that needed to be addressed.

1- The results related to the longitudinal part of the study have not been described at all. The paper claims to investigate metabotypes of newborn undergoing to therapeutic hypothermia, but the longitudinal effect due to the therapy on the metabolic signature has not been showed. If there are no results on the longitudinal part of the study, would be more appropriate to talk about metabolic signature of perinatal asphyxia in newborn.

R. We agree with the Reviewer. In the revised version of the manuscript (page 4, lines 73-78 of the “track changes” file), we have clarified that our study must be considered a hypothesis-generating study. Its main objective was to employ untargeted metabolomic analysis to identify dysregulated metabolic processes in newborns undergoing TH after PA, in an effort to establish a distinct metabotype relating to the development of pathological outcomes consistent with HIE on magnetic resonance imaging. In addition, we have tried to investigate the time evolution during TH of the discovered metabotype to evaluate the robustness of our findings and discover early putative markers of damage. We have presented the results of the longitudinal analysis based on LME modelling in section Results. Specifically, the results are summarized in Figure 2B where the effects of time and group are reported for the selected metabolites. However, the longitudinal analysis was underpowered because based on only nine patients (those with available data for all the three time points). Then, this part of the data analysis should be considered explorative and the trends discovered should be used only for hypothesis generation. We added this specification both to the results (page 14, lines 302-305) and to the limits section (page 21, lines 467-471). Lysine and L-3-methylhistidine behaved as putative markers since they showed differences both considering the first collected urines and during the whole TH.

2- I understand that collecting blood for newborn is a quite an invasive procedure. However, in the limitations of the study as well as in the text, it should be explained the rationale behind selection of urine as samples, and the limitations of these sample to investigate perinatal hypoxia.

R. Following Reviewer’s suggestion, we have specified the rationale behind the selection of urine as a biofluid, as well as the limitations of these sample to investigate perinatal hypoxia in the revised version of the manuscript. Specifically, we have added the sentence: “We selected urine as the biofluid to perform metabolomic analysis due to the possibility of non-invasive collection of the samples” in the Methods (page 5, lines 98-99), the sentence “Due to practical reasons related to the fact that asphyctic newborns do not always urinate regularly, it was not possible to collect samples at each time point for every neonate” in the Results (page 10, lines 234-236) and we have highlighted the limitation of our study “We selected urines rather than plasma as a biofluid to perform metabolomic analysis, due to the possibility of non-invasive collection of the samples. Asphyctic newborns do not always urinate regularly, and this was the principal reason for the low number of patients having urines collected at all time points” (page 20, lines 458-461). Due to ethical reasons, it was not possible to design a study with plasma sample collection. 

3- Figure 2 and 3 must be improved adding legends, name of metabolites and experimental conditions. In the way they are presented, without no information at all inside the figures, they are not so informative and also difficult to read. Probably a table would be more informative. For instance the volcano plot without name of metabolites is not really useful.

R. Thank you for this observation. We have modified figures 2 and 3 adding legends and name for the annotated metabolites (page 13, lines 276-290), in order to make them more readable and effective in integrating the information provided within the text. In particular, in figure 2 we have highlighted metabolites showing fold change greater than 2 or p-value0.05 as grey circles. We hope that in the revised version the figures result to be more informative.

4- A small discussion/hypothesis of the potential role of some of these metabolites identified with level 3 would be appropriate.

R. We agree with the Reviewer. We have improved the discussion of our results including those level 3 metabolites showing biological consistency with the clinical picture of PA. In the revised version of the manuscript, seven metabolites with Level 3 annotation have been discussed (page 18, lines 396-413; page 19, lines 414-439; page 20, lines 440-455). 

5- I do not understand the rationale behind selection of metabolites showed in figure 3. Two of these metabolites were not even mentioned in the discussion (see previous comment).

R. The four metabolites (L-lysine, L-3-methylhistidine, 2-methyl-dodecanedioic acid, S-(2-methylpropionyl)-dihydrolipoamide-E) reported in Figure 3 are metabolites that retain a significant difference between the “HIE” and “no HIE” groups during the whole TH, and not only considering the first collected urines, and may be considered putative markers. Since only L-lysine and L-3-methylhistidine have been identified (annotation with Level 1), we have reported in the conclusions only these two metabolites as putative markers (page 21, lines 476-480). Moreover, we have added a brief discussion also on the potential role of 2-methyl-dodecanedioic acid and of S-(2-methylpropionyl)-dihydrolipoamide-E (page 18, lines 412-413; page 19, lines 414-426), which have not been discussed in the first version of the manuscript due to their lower annotation level.

Reviewer #2

The manuscript is about a small scale study on infants born with Perinatal Asphyxia w/wo hypoxic-ischemic encephalopathy with the limits declared by the authors themselves. LCMS analysis followed by statistical analysis allowed to identify 111 predictors and 35 metabolites showed independent discriminatory power.

1- Of these, only four were identified ("level 1") while other 12 were annotated with a putatively less accurate "level 3". However, it is not described what this level 3 exactly means and on which basis the group of these 12 metabolites was annotated. 

R. Thank you for this observation. We have specified in the Methods section of the revised manuscript (page 10, lines 218-225 of the “track changes” file) the meaning of the annotation levels and provided the criteria for the assignment of any given level, on which basis twelve metabolites were labeled as “level 3” (ie, compounds with m/z ≤10ppm on the online databases).

2- Furthermore, two of them have been considered 'relevant metabolites' and reported in Figure 3 together with "level 1" L-Lysine and L-3-methyl histidine, i.e. 2-methyl-dodecanedioc acid and S-(2-methylpropionyl)-dihydrolipoamide E. Yet, no mention was made about these latter two metabolites supporting the observed clinical effect in reason of their putative identification. Please improve the discussion including these aspects.

R. Thank you. In the revised version of the manuscript, we have improved the discussion of our results including seven metabolites with annotation Level 3 (those showing biological consistency with the clinical picture of PA) (page 18, lines 396-413; page 19, lines 414-439; page 20, lines 440-455) and all the four metabolites reported in Figure 3 have been discussed.

3- A final consideration: if the golden window for initiating therapeutic hypothermia is in the first 6 hours of the child's life, is it realistic to imagine that in this time interval it is possible to collect the urine, carry out this kind of analysis and achieve timely results?

R. We agree with the Reviewer on the fact that this restricted time interval may pose a challenge. In the conclusions, we have speculated that our results may contribute to the future implementation of a rapid, bedside chemical test (eg, a urine dipstick) that may facilitate the early identification of newborns with PA at risk of developing brain injury (page 21, lines 480-482). Nevertheless, it could be necessary to collect urines via a catheter if the asphyctic newborn does not urinate spontaneously within six hours of life.

Minor issues:

1- Pg13 L 274 "Sixteen of these 35 variables" and not seventeen, according to Table 2

R. Thank you, we have corrected this error.

---

## [Decision Letter · Decision Letter 1]

4 Aug 2022

Urinary metabotypes of newborns with perinatal asphyxia undergoing therapeutic hypothermia

PONE-D-22-12491R1

Dear Dr. Valerio,

We’re pleased to inform you that your manuscript has been judged scientifically suitable for publication and will be formally accepted for publication once it meets all outstanding technical requirements.

Kind regards,

Andrea Motta

Academic Editor

PLOS ONE

Additional Editor Comments (optional):

Reviewers' comments:

Reviewer's Responses to Questions

**Comments to the Author**

1. If the authors have adequately addressed your comments raised in a previous round of review and you feel that this manuscript is now acceptable for publication, you may indicate that here to bypass the “Comments to the Author” section, enter your conflict of interest statement in the “Confidential to Editor” section, and submit your "Accept" recommendation.

Reviewer #1: All comments have been addressed

Reviewer #2: All comments have been addressed

2. Is the manuscript technically sound, and do the data support the conclusions?

Reviewer #1: Yes

Reviewer #2: Yes

3. Has the statistical analysis been performed appropriately and rigorously? 

Reviewer #1: Yes

Reviewer #2: Yes

4. Have the authors made all data underlying the findings in their manuscript fully available?

Reviewer #1: Yes

Reviewer #2: Yes

5. Is the manuscript presented in an intelligible fashion and written in standard English?

Reviewer #1: Yes

Reviewer #2: Yes

6. Review Comments to the Author

Reviewer #1: I would like to thanks the authors.

All my comments have been addressed and I have no further comments.

Reviewer #2: (No Response)

7. PLOS authors have the option to publish the peer review history of their article (what does this mean?). If published, this will include your full peer review and any attached files.

Reviewer #1: No

Reviewer #2: No

---

## [Editor Report · Acceptance letter]

8 Aug 2022

PONE-D-22-12491R1 

Urinary metabotypes of newborns with perinatal asphyxia undergoing therapeutic hypothermia 

Dear Dr. Valerio:

I'm pleased to inform you that your manuscript has been deemed suitable for publication in PLOS ONE. Congratulations! Your manuscript is now with our production department. 

Kind regards, 

on behalf of

Dr. Andrea Motta 

Academic Editor

PLOS ONE